# Mesophase Pitch-Based Carbon Fibers: Accelerated Stabilization of Pitch Fibers under Effective Plasma Irradiation-Assisted Modification

**DOI:** 10.3390/ma14216382

**Published:** 2021-10-25

**Authors:** Yuanshuo Peng, Ruixuan Tan, Yue Liu, Jianxiao Yang, Yanfeng Li, Jun Li, Zheqiong Fan, Kui Shi

**Affiliations:** 1Hunan Province Key Laboratory for Advanced Carbon Materials and Applied Technology, College of Materials Science and Engineering, Hunan University, Changsha 410082, China; yspeng@hnu.edu.cn (Y.P.); Tanrx@hnu.edu.cn (R.T.); LYS211301138@hnu.edu.cn (Y.L.); lyf69233@hnu.edu.cn (Y.L.); shikui@hnu.edu.cn (K.S.); 2School of Chemistry and Biological Engineering, Changsha University of Science and Technology, Changsha 410114, China; muzi.yikou@csust.edu.cn; 3School of Materials Science and Engineering, Changsha University of Science and Technology, Changsha 410114, China; fzq@csust.edu.cn

**Keywords:** mesophase pitch, carbon fiber, stabilization, plasma irradiation

## Abstract

Stabilization is the most complicated and time-consuming step in the manufacture of carbon fibers (CFs), which is important to prepare CFs with high performance. Accelerated stabilization was successfully demonstrated under effective plasma irradiation-assisted modification (PIM) of mesophase pitch fibers (PFs). The results showed that the PIM treatment could obviously introduce more oxygen-containing groups into PFs, which was remarkably efficient in shortening the stabilization time of PFs with a faster stabilization heating rate, as well as in preparing the corresponding CFs with higher performance. The obtained graphitized fiber (GF-5) from the PF-5 under PIM treatment of 5 min presented a higher tensile strength of 2.21 GPa, a higher tensile modulus of 502 GPa, and a higher thermal conductivity of 920 W/m·K compared to other GFs. Therefore, the accelerated stabilization of PFs by PIM treatment is an efficient strategy for developing low-cost pitch-based CFs with high performance.

## 1. Introduction

Carbon fibers (CFs) are widely applied in the aerospace, automotive, and wind power sectors because of their excellent mechanical and lightweight properties. CFs can be manufactured from pitch, rayon, and polyacrylonitrile precursors [1,2]. Generally, the manufacture of pitch-based CFs consists of the following processes: pitch refining, melt spinning, stabilization, carbonization, and graphitization [3]. Among the processes, stabilization has a great effect on the carbonization yield and mechanical properties of CFs [4,5]. It can prevent the fiber from melting during carbonization by promoting the fibers to form a stable oxygen bridge structure through oxidation, dehydrogenation, and crosslinking reactions [6,7]. On the other hand, the rate of oxygen diffusion from the surface toward the inside of fibers is low for oxidation and crosslinking reactions [4]. Thus, the stabilization process needs a long time and high energy, resulting in a high production cost of the corresponding CFs. It is necessary to reduce the production cost by shortening the stabilization time. However, a fast heating rate can lead to the uneven shrinkage of fibers during stabilization, causing the corresponding CFs to exhibit poor mechanical properties [8,9]. Therefore, how to shorten the stabilization time while keeping the great mechanical properties of CFs is one of the essential questions in research on the stabilization process. 

Over the last few years, it was revealed that some surface modification techniques are conducive to reducing stabilization time and promoting oxidative crosslinking reactions. Park et al. [10] introduced more oxygen into stabilized fibers (SFs) to form more reactive sites by inducing the formation of radicals through electron beam irradiation, thus reducing the stabilization time and maintaining excellent electrical conductivity and tensile strength for CFs. Morales et al. [11] induced cyclization and crosslinking of polyacrylonitrile (PAN) precursor fibers by a short UV treatment, which reduced the precursor stabilization time and improved the tensile modulus of PAN-based CFs. Furthermore, Zhao et al. [12] found that γ-ray could not only destroy the nitrile crosslinking bonds and reform nitrile groups in slight SFs, but also make additional crosslinked structures including ether crosslinking and carbon–carbon crosslinking in slight and deep SFs, which improved the thermal stability of SFs and the tensile strength of CFs. The above research showed that the surface irradiation modification of fibers is widely used in the optimization of the stabilization process.

Recently, the nonthermal plasma treatment technique has been used in surface modification due to its selective effect, low operating cost, and no pollution [13]. Wang et al. [14] used oxygen plasma to treat commercial carbon cloth, increased the surface roughness, and introduced abundant oxygen-containing functional groups to obtain the best-performing catalyst P-CC. Sim et al. [15] introduced oxygen-containing functional groups (C–O, O–C=O) to the surface of carbon fiber-reinforced plastic (CFRP) using argon and oxygen plasma treatment, which improved the wettability of the surface and adhesion between steel and CFRP. Previous studies also reported that low-temperature air plasma irradiation was an effective method to introduce a certain number of oxygen-containing functional groups on the fiber surface to give it a new structure and properties [16,17,18,19]. Therefore, our work investigated the mechanism of effective plasma irradiation-assisted modification (PIM) by analyzing the stabilization behaviors of mesophase pitch-derived pitch fibers (PFs), and we compared the properties of the corresponding CFs and graphitized fibers (GFs) under different PIM conditions, aimed at shortening the stabilization time without causing a deterioration of the physical properties of the resultant CFs and GFs.

## 2. Materials and Methods

### 2.1. Materials

Mesophase pitch with a softening point (SP) of 280 °C was prepared from fluidized catalytic cracking decant oils (FCC-DO) via two-stage reactions. The FCC-DO was first distilled at 300 °C under vacuum to remove the light components; subsequently, the vacuum residue was thermal condensed at 400 °C for 5 h under nitrogen atmosphere to obtain the spinnable mesophase pitch. As shown in Figure 1, the FCC-DO-derived mesophase pitch showed optically anisotropic features with the mesophase content reaching 100 vol.%, whereas its toluene-insoluble (TI) and quinoline-insoluble (QI) contents were measured as 70% and 51%, respectively. The pitch fibers (PFs) were prepared from the FCC-DO-derived mesophase pitch on a melt spinning apparatus equipped with a 500-hole spinneret with L/D = 0.4 mm/0.2 mm at a spinning temperature of 320 °C and nitrogen pressure of 0.5 MPa.

### 2.2. Plasma Irradiation-Assisted Modification of Pitch Fibers and Preparation of Carbon Fibers

PFs were PIM-treated in a plasma-enhanced chemical vaper deposition reactor (PECVD) with atmospheric pressure glow discharge. The conditions were set to a plasma power of 50 W and different plasma irradiation times of 0, 1, 5 and 10 min. Thereafter, the PIM-treated PFs were stabilized in a corundum tube furnace under 200 mL/min air atmosphere with a fast heating rate of 4 °C/min from room temperature to 270 °C. Then, the obtained stabilized fibers (SFs) were successively carbonized in a corundum tube furnace under 200 mL/min nitrogen atmosphere with a heating rate of 5 °C/min from room temperature to 1000 °C for 10 min. Finally, the resulting carbonized fibers (CFs) were graphitized at 2800 °C for 15 min to prepare the graphitized fibers (GFs). Therefore, the obtained fibers were labeled as PF-x, SF-x, CF-x, and GF-x, respectively, where x represents the plasma irradiation time.

### 2.3. Characterization of Pitch and Fibers

The SP of pitch was determined by a CFT-100EX capillary rheometer (Shimadzu, Kyoto, Japan). The solubility of pitch in toluene and quinoline was determined using the Soxhlet extraction method (GB/T 26930.5-2011) to obtain toluene insoluble (TI) and quinoline insoluble (QI) fractions. The optical microstructures of mesophase pitches were analyzed by using a BX53M polarized light microscopy (Olympus, Tokyo, Japan). X-ray photoelectron spectroscopy (XPS, Thermo Fisher Scientific, Waltham, MA, USA) was employed to verify the change of chemical bonds and functional groups of fibers after PIM treatment. The functional groups of fibers were also analyzed by a Nicolet iS10 Fourier transform infrared spectroscopy (FT-IR, Thermo Fisher Scientific, Waltham, MA, USA) using potassium bromide pressed-disk technique in the range of 4000–400 cm^−1^ for scanning 64 times. The oxygen (O), carbon (C), hydrogen (H) and nitrogen (N) contents of fibers were determined by a Vario EL III elemental analyzer (Elementar, Langenselbold, Germary). The oxygen content was obtained by subtraction method (O = 100 − C − H – N − S). The stabilization behaviors of PFs were characterized by a STA449 F5 thermogravimetric analyzer (TG, Netzsch, Selb, Germary) under air atmosphere with a heating rate of 5 °C/min from room temperature to 600 °C. Otherwise, thermal behaviors of SFs were characterized under argon atmosphere with a heating rate of 5 °C/min from room temperature to 1200 °C. The released gases of fibers were measured by Hiden Analytical HAS-301-1474 mass spectrometer (MS, Hiden Analytical, Warrington, UK) coupled with TG. The MS was performed at multiple ion detection (MID) mode with a secondary electron multiplier, and the quartz capillary connected to the thermal analyzer was heated to 160 °C. Raman spectroscopy (Thermo Fisher Scientific, Waltham, MA, USA) was employed to evaluate the surface crystalline size of fibers. The ratio of D-peak (1360 cm^−1^) to G-peak (1580 cm^−1^) was calculated to evaluate the structural order of fibers by using Tuinstra relationship [20]. The morphologies and diameters of both CFs and GFs were observed by JSM-6700F field emission scanning electron microscope (SEM, JEOL, Tokyo, Japan) with 5 kV. The tensile strength of CFs and GFs was measured by a XQ-1C single-filament machine (Shanghai New Fiber Instruments Co., Ltd., Shanghai, China) with a gauge length of 20 mm according to the standard (ASTM D4018-2011) from the mean value of 30 tests with the values distributing within 10%. The electrical resistivity (ρ) of GFs was measured by four probe methods with an inner and outer gauge length of 25 mm and 35 mm, respectively. The leads from the four copper wires were connected to the terminals of a Model 580 micro-ohmmeter (Keithley, Gainesville, FL, USA). The electrical resistivity (ρ) was calculated by the relationship as follows [21]:ρ = A × R/L,(1)
where R, A, and L are the electrical resistance, the cross-sectional area of the fibers, and the distance between the two potential leads, respectively.

## 3. Results and Discussion

### 3.1. Surface Chemistry of PIM-Derived Pitch Fibers

Figure 2 shows the C 1*s* XPS spectra of PFs with different PIM times. The C 1*s* peaks of PF-0 were decomposed into two peak signals at about 284.6 and 284.9 eV, as shown in Figure 2a, which were attributed to the graphitic C=C and aliphatic C–C bonds, respectively. During the PIM treatment, the oxygen free radicals attacked the aromatic rings and aliphatic groups of pitch to form oxygen functional groups on the surface of PIM-derived PFs. In comparison to pristine PFs, PIM treatment led to the appearance of a new peak signal at about 286.3 eV, which was related to the formation of C–O functional groups in PIM-derived PFs (Figure 2b–d) [22]. Table 1 summarizes the XPS results of different PIM-derived PFs using the calculation method based on the peak fitting area. It can be observed that PF-5 showed the highest oxygen content (15.4% for PF-5) compared to PF-1 and PF-10 (4.8% for PF-1 and 7.0% for PF-10). This indicates that excessive PIM treatment had a force great enough to break the chemical bonds and damage the fiber surface.

The FT-IR spectra of PIM-derived PFs are shown in Figure 3. It can be observed that PF-10 exhibited some very weak peaks at around 2960 and 3050 cm^−1^, indicating that some methyl C–H bonds and methylene C–H bonds inside aliphatic chains were broken under excessive PIM treatment. Furthermore, PF-10 showed some weak peaks at around 3100, 1260 and 1700 cm^−1^ compared to the other fibers, corresponding to –OH, C–O–C or O–C–O stretching, and carbonyl C=O stretching, respectively. This confirmed that long-term PIM treatment can result in excessive decomposition of PFs due to the plasma with high energy breaking the chemical bonds and damaging the fiber surface, which is consistent with the above XPS results. 

### 3.2. Thermal Behavior of PIM-Derived Pitch Fibers and Stabilized Fibers

In order to explore the differences in stabilization and carbonization processes of PIM-derived fibers, the PFs and SFs were analyzed by TG–MS. The stabilization weight gain of MPPF occurred due to an increase in oxygen as ketone functional groups and the loss of aliphatic carbons in pitch, and the oxygen uptake was dominant compared with the removal of low-molecular-weight components in the PFs during stabilization process [4]. Figure 4a shows the TG curves of PIM-derived PFs. It can be observed that the weight change of PF-1 behaved similarly to that of PF-0, and both the maximum weight gain and the decomposition temperature of both fibers were much lower than those of PF-5 and PF-10. This indicates that the oxidation degree of PFs was insufficient under short-term PIM treatment. Moreover, the maximum weight gain of PF-5 was higher than that of PF-10, indicating that PF-5 showed a great oxidative property in comparison with other PFs. This was also confirmed by the TG curves of SFs shown in Figure 4b. It is obvious that SF-5 showed the highest carbonization yield compared with other SFs at temperatures above 1000 °C.

The mass spectroscopy (MS) curves of PFs systematically revealed the evolution of O_2_, H_2_O, CO, and CO_2_ during the stabilization process, as shown in Figure 5. It should be noted that the primary *m*/*z* number of CO is identical to N_2_. Therefore, the *m*/*z* number of 12 as a cracking number was analyzed in multiple ion detection (MID) mode to detect the produced CO, and the intensity of the cracking number (*m*/*z* = 12) was only 5% of the intensity of the primary number (*m*/*z* = 28) [23]. In the early stage of stabilization, a certain number of ketone functional groups were introduced through oxidation and dehydrogenation, which was accompanied by the formation of H_2_O. In the later stage of stabilization, carbonyl oxygen bridge structures were formed via the insertion of oxygen. Subsequently, carbon monoxide and carbon dioxide were released through the intermolecular and intramolecular crosslinking reactions of pitch [6,24,25]. The weight uptake of PFs began at 180 °C, and PF-5 consumed the most O_2_ compared with other PFs. The H_2_O started being produced above 160 °C, and the maximum value was seen at about 270 °C. Likewise, it can be observed that the amount of H_2_O released from PF-5 was rather high compared with other PFs. The released CO and CO_2_ were derived from crosslinking reactions of pitch molecules in the late stage of stabilization, and the maximal peak values of both CO and CO_2_ were observed at about 270 °C. It is evident that the evolution trend of PF-1 was similar to that of PF-0. Moreover, the release amounts of CO and CO_2_ from PF-5 were the highest compared with other PFs. These results indicate that PIM treatment of 5 min could promote oxidation reactions during the stabilization process and form more carbonyl oxygen bridged structures. Furthermore, PF-10 released a minimal amount of CO_2_ and CO, which proves that long-term PIM treatment resulted in a decrease in the number of methyl carbons (–CH_3_) and methylene carbons (–CH_2_–) inside aliphatic chains of PFs. All results show that insufficient and excessive PIM treatment is not conducive to the introduction of oxygen functional groups in PFs.

Figure 6 shows the evolution of H_2_, CH_4_, CO_2_, and CO of SFs during their carbonization process. The H_2_ was released in the range of 600–1000 °C, and the maximum value was at about 800 °C. The evolution of CH_4_ began at 400 °C, and the maximal peak value was observed at about 600 °C, which represented the decomposition of aliphatic compounds of SFs [26]. It can be observed that the amounts of H_2_ and CH_4_ released from SF-0 and SF-1 were rather higher compared with SF-5 and SF-10. This is because the SFs obtained under short-term PIM treatment struggled to form stable crosslinked structures in the core of fibers, resulting in more H being removed at the early stage of the carbonization process due to deoxygenation [27]. CO_2_ and CO were derived from carboxyl groups and acid anhydrides obtained by oxidation, the maximal peak values of which were observed at about 500 °C and 650 °C, respectively. It is evident that the release amounts of CO_2_ and CO from SF-0 and SF-1 were far lower than from SF-5 or SF-10. Moreover, there is no doubt that the release amounts of CO_2_ and CO from SF-5 were the highest compared with other SFs due to the increase in oxygen-containing functional groups of SF-5.

Moreover, the elemental analysis and C/O ratio of different PIM-derived PFs and SFs are summarized in Table 2. It can be observed that PF-0 contained 0.78 wt.% oxygen with a C/O ratio of 161.6. The oxygen content and C/O ratio of PF-5 were found to be 2.95 wt.% and 41.6, strongly indicating that the appropriate PIM treatment is conducive to the introduction of oxygen. In addition, PIM-derived SFs behaved similarly to the elemental analysis results of PIM-derived PFs. SF-5 had the highest oxygen content of 12.52 wt.% and lowest C/O ratio of 8.9, which also indicates that PF-5 had a great oxidative property in forming a stable crosslinked structure in SF-5, promoting the resultant stabilization process.

### 3.3. Microstructure and Properties of PIM-Derived Carbonized Fibers and Graphitized Fibers

Figure 7 shows the Raman spectra of different PIM-derived PFs, SFs, CFs, and GFs. The Raman D-band at around 1360 cm^−1^ and G-band at around 1580 cm^−1^ correspond to the defect lattice vibration mode and ideal graphite lattice vibration mode, respectively [28]. Therefore, the intensity ratio of the D-band and G-band (I_D_/I_G_) allows evaluating the carbon crystallite character, and a lower value of I_D_/I_G_ indicates fewer defect sites and more highly ordered orientation in fibers [29]. It is evident that the I_D_/I_G_ ratios of both PF-5 and SF-5 were much higher than those of other PFs and SFs (I_D_/I_G_ = 0.89 for PF-5, I_D_/I_G_ = 0.94 for SF-5), related to their oxygen crosslinked structure with the highest oxygen content, as shown in Table 2. However, the I_D_/I_G_ ratios of both CF-5 and GF-5 (I_D_/I_G_ = 1.09 for CF-5, I_D_/I_G_ = 0.09 for GF-5) were much lower than those of other CFs and GFs, related to fewer defect structures and their highly ordered carbon crystallite orientation, in accordance with their higher mechanical properties and thermal conductivity.

Figure 8 shows the cross-section SEM images of different PIM-derived CFs and GFs. It can be observed that there were significant changes in the cross-section of CFs after graphitization The cross-section structure of CFs and GFs was also improved with the increase in PIM time. The cross-section of CF-0 appeared as a typical skin–core structure, and the corresponding GF-0 had a significant hollow structure due to the deterioration of the core structure with insufficient oxidation degree during the graphitization process. The skin–core structure of CF-1 was weakened, and the corresponding GF-1 revealed several micropores. Moreover, the cross-sections of CF-10 and GF-10 showed a cracked structure and hollow structure, respectively. However, it is worth noting that CF-5 and GF-5 showed fewer defects and better carbon layer orientation compared to other fibers, respectively. These views are consistent with the results of Raman analysis. Moreover, Table 3 summarizes the main properties of different PIM-derived CFs and GFs. It is evident that CF-0, CF-1, and CF-10 showed a poor tensile strength and tensile modulus, and the corresponding GF-0, GF-1, and GF-10 exhibited low mechanical properties and thermal conductivities, related to their skin–core, cracked, microporous, and hollow structures. In contrast, CF-5 showed optimal mechanical properties (tensile strength of 1.34 GPa, tensile modulus of 121 GPa), and the corresponding GF-5 also had the best mechanical properties (2.21 GPa for tensile strength, 502 GPa for tensile modulus) and the highest thermal conductivity (λ) of 920 W/m·K. These results suggest that moderate PIM treatment (PIM time of 5 min in this experiment) of PFs can optimize the stabilization behaviors of PFs by accelerating the oxidation and crosslinking processes. Consequently, the corresponding CFs and GFs showed excellent properties and microstructure.

## 4. Conclusions

The stabilization and carbonization behaviors of MP-derived PFs under effective PIM treatment were systematically investigated. This work demonstrated that PIM treatment could promote the oxidation crosslinking reaction of PIM-derived PFs via the introduction of oxygen-containing polar groups into pitch. However, insufficient and excessive PIM conditions were not conducive to generating ideal microstructures and properties of the corresponding CFs and GFs. Herein, when the PIM time was 5 min, the obtained CF-5 and GF-5 showed the best mechanical properties and an optimal thermal conductivity compared to other corresponding fibers. Therefore, this study can help us further optimize the stabilization process of PFs and achieve the target for reducing the production cost of pitch-based CFs through accelerated stabilization based on PIM treatment.

## Figures and Tables

**Figure 1 materials-14-06382-f001:**
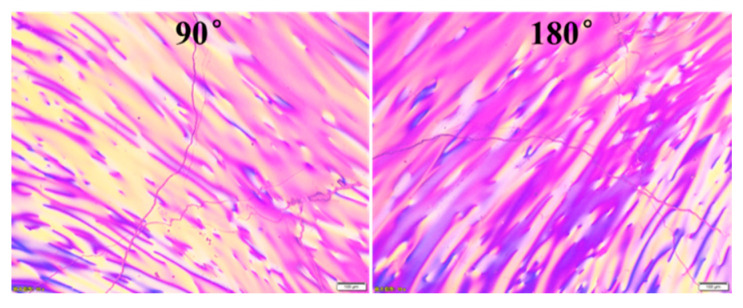
Optical micrographs of FCC-DO mesophase pitch.

**Figure 2 materials-14-06382-f002:**
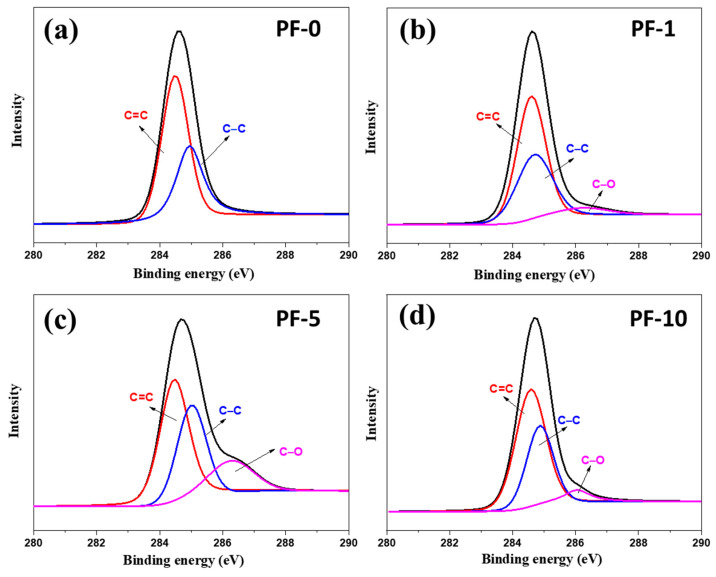
C 1*s* XPS spectrum of different PIM-derived pitch fibers: (**a**) PF-0; (**b**) PF-1; (**c**) PF-5; (**d**) PF-10.

**Figure 3 materials-14-06382-f003:**
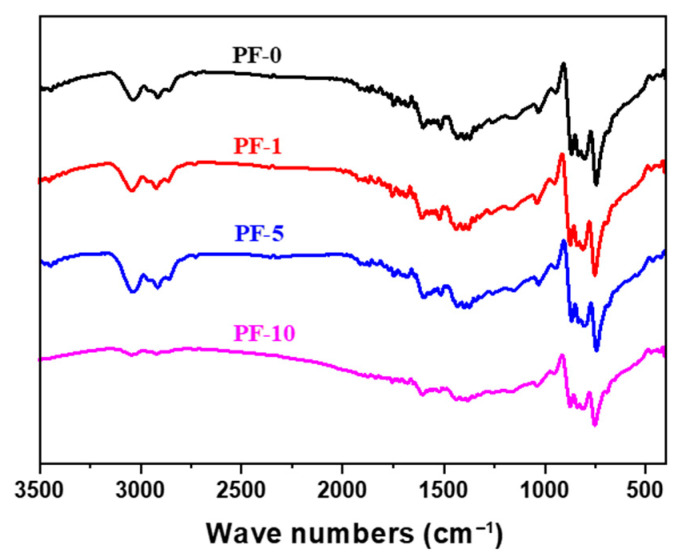
FT-IR spectra of PIM-derived pitch fibers.

**Figure 4 materials-14-06382-f004:**
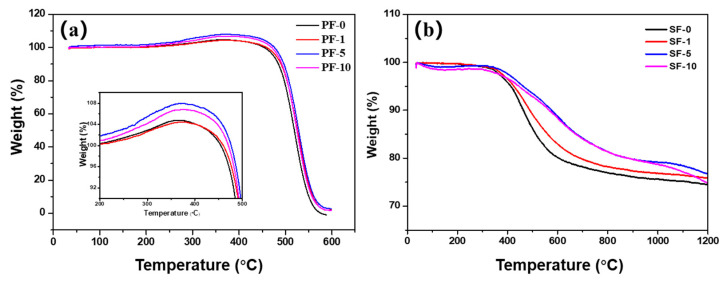
TG curves of different PIM-derived (**a**) pitch fibers and (**b**) stabilized fibers.

**Figure 5 materials-14-06382-f005:**
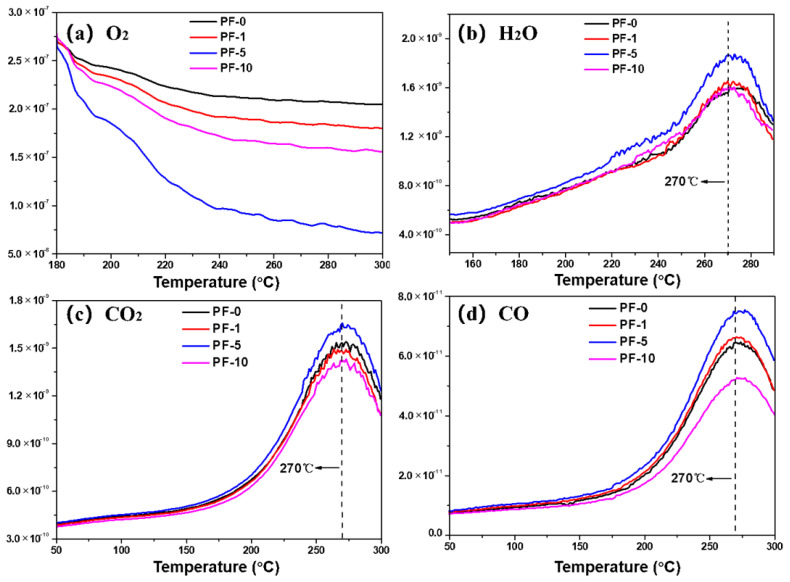
MS curves of different PIM-derived pitch fibers in air: (**a**) O_2_; (**b**) H_2_O; (**c**) CO_2_; (**d**) CO.

**Figure 6 materials-14-06382-f006:**
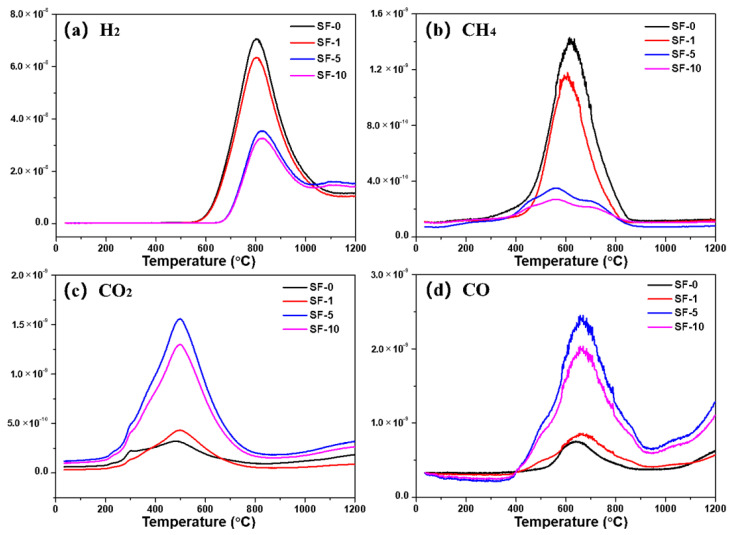
MS curves of different PIM-derived stabilized fibers in argon: (**a**) H_2_; (**b**) CH_4_; (**c**) CO_2_; (**d**) CO.

**Figure 7 materials-14-06382-f007:**
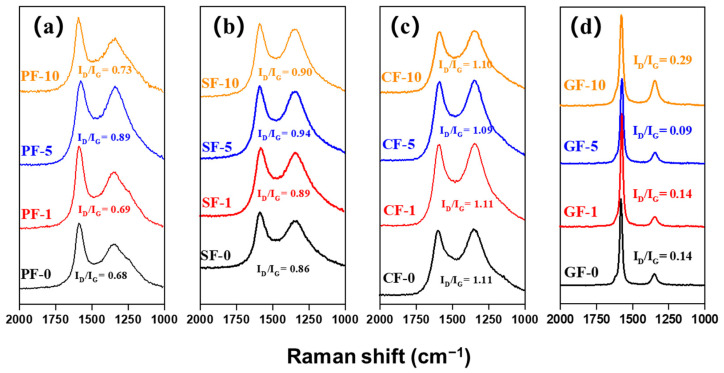
Raman spectrum of different PIM-derived (**a**) pitch fibers, (**b**) stabilized fibers, (**c**) carbonized fibers and (**d**) graphitized fibers.

**Figure 8 materials-14-06382-f008:**
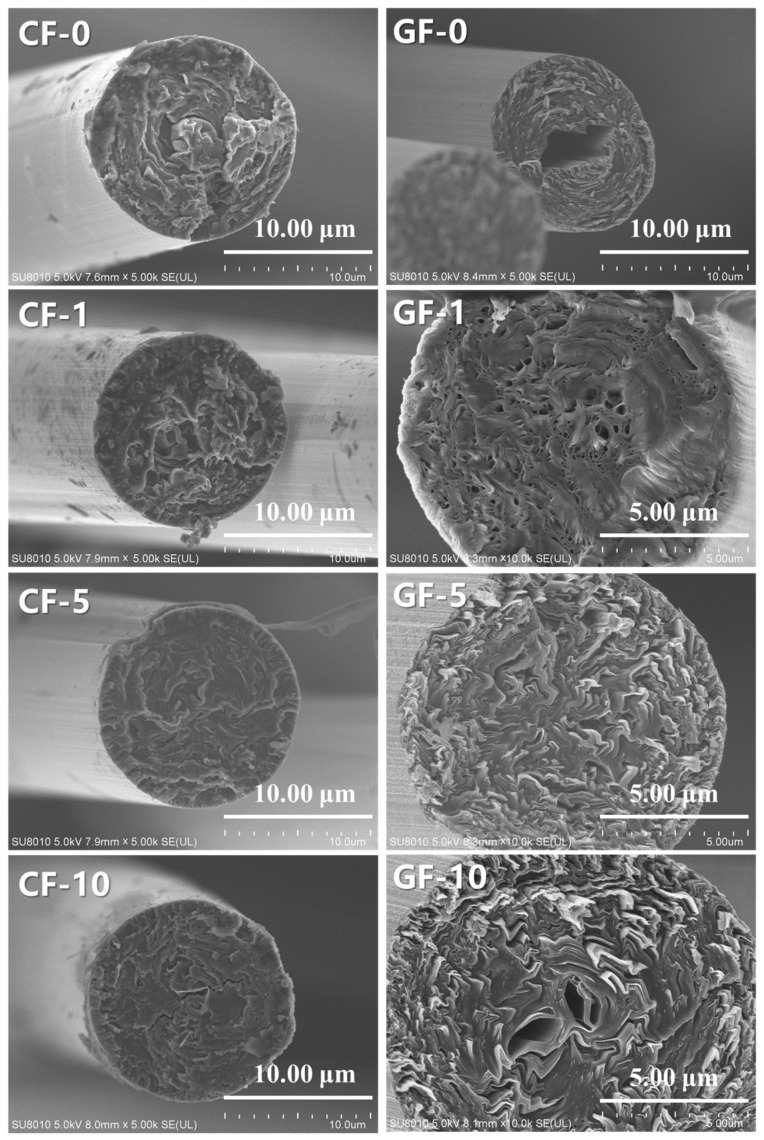
SEM images of different PIM-derived carbonized fibers and graphitized fibers.

**Table 1 materials-14-06382-t001:** XPS Results of Different PIM-Derived Pitch Fibers.

Samples	C 1*s* Functional Group (%)			XPS Analysis (at.%)	
	C=C	C–C	C–O	C	O
PF-0	67.8	32.2	-	96.65	3.35
PF-1	60.2	35.0	4.8	94.27	5.73
PF-5	45.4	39.2	15.4	83.33	16.67
PF-10	53.2	39.8	7.0	91.73	8.27

**Table 2 materials-14-06382-t002:** Elemental Analysis and C/O Ratio of Different PIM-Derived Pitch Fibers and Stabilized fibers.

Samples	C (%)	H (%)	N (%)	O (%) ^a^	C/O
PF-0	94.57	4.39	0.26	0.78	161.6
PF-1	94.41	4.26	0.25	1.08	116.5
PF-5	92.03	3.77	0.25	2.95	41.6
PF-10	94.26	3.27	0.25	2.22	56.7
SF-0	87.24	2.87	0.25	9.64	12.1
SF-1	85.36	3.41	0.39	10.84	10.5
SF-5	83.60	3.55	0.33	12.52	8.9
SF-10	87.09	2.79	0.13	9.99	11.6

^a^ Oxygen content was calculated by difference.

**Table 3 materials-14-06382-t003:** Main Properties of Carbon Fibers.

Samples	Elongation (%)	TS (GPa)	TM (GPa)	ρ (μΩ·m)	λ ^a^ (W m^−^^1^ K^−1^)
CF-0	0.86	0.49	57	-	-
CF-1	0.89	0.72	81	-	-
CF-5	1.11	1.34	121	-	-
CF-10	0.94	0.88	94	-	-
GF-0	0.43	0.89	207	3.02	418
GF-1	0.45	1.37	304	2.73	462
GF-5	0.44	2.21	502	1.37	920
GF-10	0.47	1.76	374	1.82	693

^a^ Axial thermal conductivity calculated using the following formula: λ = 1261/ρ.

## Data Availability

All data included in this study are available upon request by contact with the corresponding author.

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
