# Peer review of "Mesophase Pitch-Based Carbon Fibers: Accelerated Stabilization of Pitch Fibers under Effective Plasma Irradiation-Assisted Modification"

_materials, 2021, doi:10.3390/ma14216382_

Round 1

Reviewer 1 Report

This manuscript presents an interesting work on investigation of mechanism of oxygen plasma stabilization and change in microstructure depending on oxygen content. Many analysis including EA, XPS, SEM, Raman spectrum and MS have been conducted. However, this manuscript should be revised for submission to materials; the reasons of this are listed as below.

  1. In abstract ‘a most’ should be modified as ‘the most’.
  2. Your English needs some proofreading.
  3. In section 2.2, I wonder what PECVD means. You need to figure out what the abbreviation means.
  4. In section 2.3, when the softening point is described as SP, it is necessary to describe the softening point as SP in section 2.1 like softening point(SP)
  5. In page 5 line 170, I wonder what MID mode. You need to figure out what the abbreviation means
  6. In XPS and EA results, why the C/O ratio is derived differently?
  7. In section 2.1 and 2.2, there is a reference to polarizing microscope analysis, but it is not presented in the results section. The results of the polarizing microscope analysis should be presented.

Author Response

Response to Reviewer 1:

 We would like to express our sincere thanks to this reviewer for her/his positive evaluation, valuable comments and constructive suggestions.

(General comment)

This manuscript presents an interesting work on investigation of mechanism of oxygen plasma stabilization and change in microstructure depending on oxygen content. Many analysis including EA, XPS, SEM, Raman spectrum and MS have been conducted. However, this manuscript should be revised for submission to materials; the reasons of this are listed as below.

(Answer to General comment)

Thank you very much for kind recommendation. We have tried our best to revise the manuscript according to your valuable comments and we truly hope that this work could be published in Materials.

(Comment 1)

In abstract ‘a most’ should be modified as ‘the most’.

(Answer to Comment 1)

Thank you very much for kind comments. We are truly sorry for our mistakes about the manuscript. We have revised the mistake. (Page 1)

(Comment 2)

Your English needs some proofreading.

(Answer to comment 2)

Thank you for valuable comment. We have had the manuscript checked by a native English-speaking colleague.

 (Comment 3)

In section 2.2, I wonder what PECVD means. You need to figure out what the abbreviation means.

(Answer to comment 3)

Thank you very much for kind comment. The PECVD stands for plasma enhanced chemical vaper deposition. We have added it in section 2.2. (Page 2)

(Comment 4)

In section 2.3, when the softening point is described as SP, it is necessary to describe the softening point as SP in section 2.1 like softening point (SP).

(Answer to comment 4)

Thank you very much for good suggestion. We have added the information according to your suggestion. (Page 2)  

(Comment 5)

In page 5 line 170, I wonder what MID mode. You need to figure out what the abbreviation means.

(Answer to comment 5)

Thank you very much for kind comment. We have supplied the meaning of MID mode (multiple ion detection mode). (Page 5)

(Comment 6)

In XPS and EA results, why the C/O ratio is derived differently?

 (Answer to comment 6)

Thank you for valuable comment. XPS is a semi-quantitative technique for surface chemical analysis allowing obtaining an atomic percentage of the various elements present in the analyzed specimens as well as information about atomic environment of each element. The strength of photoelectrons is not only related to the concentration of atoms, but also to the average free path of photoelectrons, the surface finish of samples, the chemical state of elements, the strength of X-ray sources and the state of instruments. Therefore, XPS technology generally cannot provide the absolute content of the analyzed elements, but only the relative content of each element. Therefore, the C/O ratio is derived differently in XPS and EA results.

(Comment 7)

In section 2.1 and 2.2, there is a reference to polarizing microscope analysis, but it is not presented in the results section. The results of the polarizing microscope analysis should be presented.

(Answer to comment 7)

Thank you for your kind suggestion. We have supplied the results of the polarizing microscope analysis in section 2.1. (Page 2)

Reviewer 2 Report

1. Indicate how many hours the graphitization was maintained at 2800°C during the production of GFs in line 93.

2. In the XPS analysis in Section 3.1, how might C-O functional group formation affect PF with increasing plasma treatment time?

Also, what is the basis for calculating the at% of C and O using XPS? The calculation method should be presented.

3. Figure 3 shows the TG curves of PIM-derived PFs at line156. This sentence is for Figure 3(a), so it should be corrected.

4. In Figure 3(a), add a sufficient reason for the weight increase in the 200~500℃ section to the text.

5. In the SEM image of Figure 7, hollows that could not be observed in CF-0 were observed in GF-0 after graphitization. Further explanation is needed in the content by what mechanism the hollow was formed during graphitization.

Author Response

Response to Reviewer 2:

We would like to express our sincere thanks for valuable comments and constructive suggestions.

(Comment 1)

Indicate how many hours the graphitization was maintained at 2800°C during the production of GFs in line 93.

(Answer to comment 1)

Thank you very much for kind comment. We have added the information according to your suggestion. (Page 3)

(Comment 2)

In the XPS analysis in Section 3.1, how might C-O functional group formation affect PF with increasing plasma treatment time? Also, what is the basis for calculating the at% of C and O using XPS? The calculation method should be presented.

 (Answer to comment 2)

Thank you very much for kind comment. During the PIM treatment, the oxygen free radicals attacked the aromatic rings and aliphatic groups of pitch molecular to form the oxygen functional groups on the surface of PFs. The at% of C and O, C1s functional groups percentages were calculated by the fitting area of each group. We have added the discussion and the calculation method in the revision. (Page 3)

(Comment 3)

Figure 3 shows the TG curves of PIM-derived PFs at line156. This sentence is for Figure 3(a), so it should be corrected.

 (Answer to comment 3)

Thank you for valuable comment. We are truly sorry for the vagueness of our expression. And we have corrected the mistake. (Page 5)

(Comment 4)

In Figure 3(a), add a sufficient reason for the weight increase in the 200~500 section to the text.

(Answer to comment 4)

Thank you for valuable comment. The weight gain of PFs occurs due to an increase in oxygen as ketone functional groups and the loss of aliphatic carbons in pitch. In fact, the weight gain of PFs due to oxygen uptake was dominant compared with the removal of low molecular weight components in the PFs during stabilization process This conclusion has been confirmed in reference [4]. We have added the reasonable description based on your suggestion in the revision. (Page 5)

 (Comment 5)

In the SEM image of Figure 7, hollows that could not be observed in CF-0 were observed in GF-0 after graphitization. Further explanation is needed in the content by what mechanism the hollow was formed during graphitization.

(Answer to comment 5)

Thank you very much for kind comment. The cross-section of CF-0 appeared a typical skin-core structure, and the core structure with insufficient oxidation degree would be deteriorated into holes on the GF-0 after graphitization process. We have added the corresponding description in the revision. (Page 8)

Reviewer 3 Report

The authors describe a plasma based method to improve carbon fibers made from pitch fibers. While this seems to be an interesting study there are some points that neeed to be adressed:

1) All the acronyms used get me confused. I would write the words carbon fibers, graphitized fibers, etc. without the acronyms. I think that would make it much easier for the reader.

2) What is SF in line 46? It is not explained in the text.

3) What is CFRP in line 63?

4) While I happen to know what PECVD means, it still needs to be expklained in the text.

5) On page 4 it is shown that the amount of oxygen decreases with the longest plasma treatment. I do not understand why this is the case. In a oxygen containing plasma this is highly surprising and needs to be discussed in more detail.

6) In the FTIR the OH and CH vibrations around 3000 cm-1 get reduced. However the peaks around 1500 cm-1 seem much less influenced. That would suggest a similar amout of C=O containing species.

7) MS stands for mass spectroscopy I assume (line 176).

8) How were the cross sections of the fibers in the SEM cut? Could some of the features seen in the SEM stem from cutting?

Author Response

Response to Reviewer 3:

We would like to express our sincere thanks for valuable comments and constructive suggestions.

(General comment)

The authors describe a plasma based method to improve carbon fibers made from pitch fibers. While this seems to be an interesting study there are some points that need to be addressed.

(Answer to General comment)

Thank you very much for kind recommendation. We have tried our best to revise the manuscript according to your valuable comments and we truly hope that this work could be published in Materials.

(Comment 1)

All the acronyms used get me confused. I would write the words carbon fibers, graphitized fibers, etc. without the acronyms. I think that would make it much easier for the reader.

(Answer to comment 1)

Thank you for valuable comment. We apologize for our vague expressions. However, the CFs and GFs was easily understood as carbon fibers and graphitized fibers in the field of carbon materials. Hope you can understand our descriptions.

(Comment 2)

What is SF in line 46? It is not explained in the text.

 (Answer to comment 2)

Thank you very much for kind comment. The SFs stands for stabilized fibers. We have added it in the text. (Page 1)

(Comment 3)

What is CFRP in line 63?

 (Answer to comment 3)

Thank you for valuable comment. We are truly sorry for the vagueness of our expression. CFRP means the carbon fiber reinforced plastics. We have added it in the text. (Page 2)

 (Comment 4)

While I happen to know what PECVD means, it still needs to be explained in the text.

(Answer to comment 4)

Thank you very much for kind comment. The PECVD stands for plasma enhanced chemical vaper deposition. We have added it in section 2.2. (Page 2)

 (Comment 5)

On page 4 it is shown that the amount of oxygen decreases with the longest plasma treatment. I do not understand why this is the case. In an oxygen containing plasma this is highly surprising and needs to be discussed in more detail.

(Answer to comment 5)

Thank you very much for kind comment. It confirmed that the long-time PIM treatment would result in excessive decomposition of PFs due to the plasm with high energy would break the chemical bonds and damage the fiber surface, which was consistent with the above XPS results. We have added the reason in the manuscript. (Page 4)

 (Comment 6)

In the FTIR the OH and CH vibrations around 3000 cm-1 get reduced. However, the peaks around 1500 cm-1 seem much less influenced. That would suggest a similar amount of C=O containing species.

(Answer to comment 6)

Thank you very much for your great comment. We totally agreed with your opinion on the FTIR results. We have detailly explained the result in the revision. (Page 4) 

 (Comment 7)

MS stands for mass spectroscopy I assume (line 176).

(Answer to comment 7)

Thank you very much for kind comment. MS stands for mass spectroscopy. We have mentioned it in section 2.3. (Page 3)

 (Comment 8)

How were the cross sections of the fibers in the SEM cut? Could some of the features seen in the SEM stem from cutting?

(Answer to comment 8)

Thank you very much for kind comment. In this experiment, we cut the cross sections of the fibers frozen by liquid nitrogen with scissors. Therefore, the cross-section is very flat.
